# Towards Early Detection of Tropospheric Aerosol Layers Using Monitoring with Ceilometer, Photometer, and Air Mass Trajectories

**Mariana Adam** [1,*] **, Konstantinos Fragkos** [1] **, Ioannis Binietoglou** [2] **, Dongxiang Wang** [3,4] **, Iwona S. Stachlewska** [3] **, Livio Belegante** [1] **and Victor Nicolae** [1]

[1] National Institute of Research and Development for Optoelectronics—INOE 2000, 077125 Magurele, Romania; kostas.fragkos@inoe.ro (K.F.); livio@inoe.ro (L.B.); victor.nicolae@inoe.ro (V.N.)

[2] Institute for Astronomy, Astrophysics, Space Applications and Remote Sensing, National Observatory of Athens, GR-15 236 Athens, Greece; binietoglou@noa.gr

[3] Faculty of Physics, University of Warsaw, 02-093 Warsaw, Poland; dongxiang.wang@fuw.edu.pl (D.W.); iwona.stachlewska@fuw.edu.pl (I.S.S.)

[4] SEPCOIII Electric Power Construction Co., Ltd., Qingdao 266100, China

[*] Correspondence: mariana.adam@inoe.ro; Tel.: +40-21-4574522

**Abstract:** A near-real-time automatic detection system, based on the synergy of continuous measurements taken by a ceilometer and a photometer, has been implemented in order to detect lofted atmospheric aerosol layers and estimate the aerosol load. When heavy-loaded conditions are detected (defined by a significant deviation of the optical properties from a 10-year climatology), obtained for aerosol layers above 2500 m, an automatic alert is sent to scientists of the Romanian Lidar Network (ROLINET) to further monitor the event. The Hybrid Single-Particle Lagrangian Integrated Trajectory (HYSPLIT) back-trajectory calculations are used to establish the possible pollution source. The aerosol transport events are considered to be major when various optical properties provided by the photometer are found outside the climatological values. The aerosol types over the three years for all the events identified revealed that the contribution to the pollution was 31%, 9%, and 60% from marine, dust, and continental types. Considering only the 'outside climatology limits' events, the respective contribution was 15%, 12%, and 73% for marine, dust, and continental types, respectively.

**Keywords:** ceilometer; photometer; back-trajectories; near-real-time monitoring; early detection system

## 1. Introduction

Atmospheric particulate pollution can impact human health and safety and, to a larger extent, the biosphere [1–4]. Heavily loaded aerosol layers in troposphere can affect aviation [5], and when reaching the ground, they can diminish air quality [6], while they also affect agriculture areas or important infrastructure [6]. For these reasons, the continuous monitoring of atmospheric particulate matter (PM) is of importance. Extended networks of surface air pollution tracking have been developed (e.g., EEA in Europe, EPA in US, etc.). They continuously measure the PM levels using in situ instrumentation and often circulate alerts when atmospheric pollution levels exceed specific limits. The aerosol layers in free troposphere (FT) can be considered as a proxy for their potential to enter PBL and, thus, can directly affect human activity. On the other hand, it is a reliable dataset when used for the validation of the aerosol layer height (ALH) retrieved from satellites (e.g., [7,8]) and of chemistry transport models [9]. Furthermore, they can provide a warning for aviation (e.g., [10]). The aim of the current study is to introduce a system of aerosol early detection which has also the potential to be extended for the monitoring of the free troposphere aerosol distribution. This monitoring is very important as it can give information about changes in the atmospheric circulation that affect the aerosol distribution and, consequently, the aerosol radiative effects [11].

The monitoring of PM in the free troposphere is mostly based on active remote sensing techniques. Lidars have been used extensively in the recent decades to monitor atmospheric pollution in the free troposphere, while lidar networks, such as EARLINET [12], MPLNET [13], LALINET [14], NDACC [15], and AD-Net [16], are implemented to monitor the pollution extent over large regions. The list of acronyms is given in Abbreviations. Various types of aerosol layers above the boundary layer have been studied, such as volcanic ash (e.g., [17–21]), biomass burning (e.g., [4,22–30]), or dust (e.g., [16,25,31–36]). In general, the lidar networks do not run continuously, mainly due to the high running and maintenance costs and the human resources involved (e.g., [37,38]). Over the last decade, ceilometers, primarily designed to monitor cloud layers, have been considered for aerosol monitoring while large networks, such as automated lidars and ceilometers (ALCs) in E-PROFILE [39], have been established. Compared to multiwavelength lidars, ceilometers use a single laser wavelength, typically in the NIR region, while their laser power is smaller. Thus, the retrieval of particle backscatter coefficient is not trivial for ceilometers due to the low signal to noise ratio (SNR). However, ceilometers, opposed to lidars, have few advantages that make them very appealing for pollution monitoring: continuous unattended operation and lower running cost. Two key manufacturers of the ceilometers are Lufft [40] and Vaisala [41]. The Lufft ceilometer uses a laser emitting light at 1064 nm (a typical wavelength found in Nd-YAG based high-power lidars) with a power of ~50 mW and photon counting detection. Vaisala ceilometers typically emit light at 910 nm (which needs correction for water vapor absorption) with the analog detection, and a power of ~12 mW. The potential of the Lufft ceilometers to provide the aerosol backscatter coefficient was shown by Heese et al. [42], among others, where comparisons with lidar retrievals were performed. The authors report SNR > 1 up to 4–5 km during the day and up to 8.5. km at night. Wiegner et al. [9] report on the reliable detection of the elevated layers up to 5 km. For Lufft ceilometers, the retrieval of the aerosol backscatter coefficient is based on a molecular (Rayleigh) calibration, similar with the typical lidar procedure [9]. For Vaisala ceilometers, the calibration is based on clouds [43].

The near-real-time (NRT) retrievals of aerosol backscatter coefficient, constrained using the aerosol optical depth (AOD) provided by photometer operated within AERONET [44], have been implemented within the Iberian ceilometer network [45]. In that study, the ceilometers and the photometers in the AERONET network are manufactured by Lufft [40] and Cimel Electronique [46], respectively. Traditionally, ceilometer networks were established within National Meteorological or Environmental Services (e.g., Met Office, UK; DWD, Germany; NWS, US) for cloud monitoring for aviation purposes and, lately, to improve weather forecasts. Wiegner et al. [9] showed that the ceilometers can be reliable in determining the aerosol layers up to 5 km. Observations of a dust event in lower troposphere in the Gobi Desert are also reported by Kawai et al. [47]. Marcos et al. [48] reported on analysis of aerosol backscatter coefficient from CL51 ceilometer by Vaisala [41] over four years of observations. Based on comparisons with collocated lidar, the authors found that the ceilometer is able to identify the aerosol layers while the backscatter retrieval is overestimated at high altitudes (over 3 km) by 13% as compared with lidar retrievals. The performance of three different ceilometers is presented by Madonna et al. [49]. The authors report that the attenuated backscatter coefficient (based on comparisons with the retrieval from a Raman lidar) provided by three ceilometers (CHM15k, CS135s, and CT25K) reveals differences due to different signal-to-noise ratios, but also due to changes in ambient temperature which affects the stability of the ceilometer calibration. Several studies of the planetary boundary layer (PBL) were reported, investigating either the PBL height (PBLH) or the PBL structure [50–57] based on ceilometer data.

As mentioned before, NRT monitoring of pollution events is crucial for their early detection, evaluation, and possible alerting if adverse impacts are expected. For example, the nine Volcanic Ash Advisory Centres (VAACs) monitor volcanic ash. VAAC London uses a ceilometer network for NRT monitoring supervised by VAAC forecasters [58–61]. These kinds of NRT alert systems are not only limited on pollution assessment, but they

can also be used for any other environmental hazards. For example, Haeffelin et al. [62] set up an alert system for radiation fog which could be implemented in near real time in ALC stations. Papagiannopoulos et al. [10] demonstrated an early warning (in NRT) system for aerosol aviation hazards using a lidar with depolarization on two occasions, namely a dust intrusion in March 2019 and a volcanic ash intrusion in June 2019. In [10], the warning is based on the estimation the backscatter coefficient which exceeds a threshold, where the whole profile is considered (i.e., no layers are particularly determined). Some advanced tools for profiling the atmosphere by use of high-power lidars data (usually multiwavelength information) are limited in terms of their use in NRT operational mode (e.g., Grasp/GARRLiC, https://www.grasp-open.com/, last access 24 August 2021, [63]). Thus, we consider it not adequate for our goal at this stage. Similarly, there are methods described to assess PBLH; however, they are not implemented in NRT.

Within this context, we implemented an automated system to detect free troposphere pollution layers in NRT, where the degree of pollution is evaluated based on photometer measurements. When the pollution is high, we issue an automated alert towards designated scientists for further thorough monitoring and analysis. The procedure is easy to implement in any location where a lidar or ceilometer and a photometer are available (e.g., MPLNET locations). One secondary goal of this study is to exploit the information provided by the ceilometer' manufacturer, given in the raw data. We refer to ALH, which is given as 'pbl' in the raw data, while the manual describes 'pbl' as aerosol layers. Here, we present the methodology of setting the early detection system implemented at Magurele, Romania (WMO Integrated Global Observing System id: 0-20008-0-INO). The first results were presented during the 29th ILRC conference [64]. Section 2 presents the resources and the methodology while Section 3 shows example results, along with a discussion. We conclude the paper in Section 4.

## 2. Resources and Methodology

### 2.1. CHM15k Ceilometer Data

During February 2018, two Lufft CHM15k ceilometers [40] were installed at INOE 2000 facilities, at two close locations (~1.5 km apart): the Romanian Atmospheric 3D research Observatory (RADO) [65] and the Magurele Centre for Atmosphere and Radiation Studies (MARS) [66]. Both ceilometers are currently part of E-PROFILE infrastructure [39], while one is also part of CloudNet within the pan-European Aerosol, Clouds, and Trace Gases Research Infrastructure (ACTRIS) [67]. The instruments operate at 1064 nm, with a pulse energy of 7 μJ and a pulse repetition frequency of 5–7 kHz. The field of view of the receiver is 0.45 mrad, while the full overlap is around 1.5 km. An overlap correction is applied; thus, the signal down to 200 m is used (correction function provided by manufacturer). The current study uses the data from the ceilometer with serial number CHM170137, located on MARS site (CHM15k 0-20008-0-INO A in E-PROFILE and CloudNet). The ceilometer provides quick looks of range-corrected signals (RCS) in NRT (updates are set to 15 min), which are visualized at a dedicated public webpage [68].

For this study, we use the ceilometer data to detect the presence of lofted aerosol layers in the atmosphere and determine ALH. Note that ALH, the altitude of the pollution layers, is taken as the top of the pollution layer (as provided in the raw ceilometer data by the manufacturer's algorithm). Currently, we use the second and the third ALH, since the first ALH usually corresponds to PBLH. Note that the ALH provided by the ceilometer are limited to 4–5 km altitude; thus, this early detection system does not currently cover the upper troposphere. The development of an in-house algorithm to determine the aerosol layers and improve the vertical extent of the early detection system is ongoing [23].

### 2.2. Photometer Data

The sun photometer measurements started at INOE in 2007 with the installation of a sun/sky CIMEL Electronique 318A (referred as C318A) spectral radiometer (data can be found at AERONET website under the station "Bucharest_INOE"). Since 2015,

a new AERONET [44] station has been developed following the purchase of a Cimel CE318-T (referred as C318T) sun/sky/lunar (triple) photometer [44] (AERONET station ID: "Magurele_INOE"). The two instruments were operated side by side until May 2016, when the C318A was relocated to Remote Sensing Laboratory of UW in Warsaw, Poland. The Cimel Electronique instrument is a multiband sun photometer that takes spectral measurements of the direct sun irradiance and the sky radiance in the almucantar and principal plane configurations using interference filters [69]. The filters are centered at the nominal wavelengths of 340, 380, 440, 500, 675, 870, 1020, and 1640 nm, while an additional channel at 935 nm is used to retrieve the total precipitable water [69]. The aerosol optical depth (AOD) can be obtained from the direct sun observations through the Beer–Lambert law [69], while the Ångström exponent (AE), fine-mode (FMAOD), and coarse-mode (CMAOD) AOD at 500 nm and fine-mode fraction (FMF) are retrieved using the ONEILL algorithm [70]. In addition, the photometer performs sky radiance measurements at the nominal wavelengths of 440, 675, 870, and 1020 nm. The microphysical aerosol properties can be obtained from the inversion of the sky radiance measurements [71,72]. Carstea et al. [73] analyzed ten years (2007–2016) of C318A data to obtain the AOD climatology and their long-term changes over Magurele. The following variables were analyzed: AOD (at 340 nm, 500 nm, 870 nm, and 1020 nm); AE (at 440–870 nm); and fine-mode FMAOD, CMAOD, and FMF (at 500 nm). The mean monthly values reported will be considered in the methodology section. As the current study started in 2018, we considered that the climatology until 2016 was good enough. For consistency, we use the same climatology for the entire analyzed period. The current photometer data are obtained from the C318T photometer and refer to Version 3, Level 1.5 [74]. Level 1.5 data are cloud-screened and quality controlled, but not quality-assured like Level 2 data, since the two consecutive calibrations of the instrument have not been verified. Since this study is focused in a NRT implementation, Level 2 data are not available.

### 2.3. HYSPLIT Model

The HYSPLIT back-trajectory model [75] is used to identify the possible source of the aerosol pollution layers. The input data in the model represent the ALH provided by ceilometer. The simulations are based on GFS0.25 meteorological data (since June 2019), while lower resolution data (e.g., GDAS0.5 or GDAS1) are used in case the former are not available. One backward trajectory is calculated for each detected aerosol layer (ASL) for a backward run time of 240 h. The 10-day period was selected in line with what the community uses in order to identify the source of long-range transported aerosol layers [25,26]. All the output meteorological variables are saved in text files for further processing.

### 2.4. Methodology

The steps followed in the methodology proposed are presented in Figure 1 and discussed below. In the first step, the quick looks of RCS are updated and shown on the webpage every 15 min for plots, representing the last 24 h of measurements. We superimpose the cloud base height (CBH) and the ALH over RCS images. The raw data files save 5 min measurements taken at 30 s (10 profiles) and 15 m temporal and spatial resolutions (as required for E-PROFILE). We check if we have ALH (second and third ALH recorded in the raw file) above 2500 m altitude a.g.l. in the last 15 min. The PBLH in Magurele is usually below 2500 m [76].

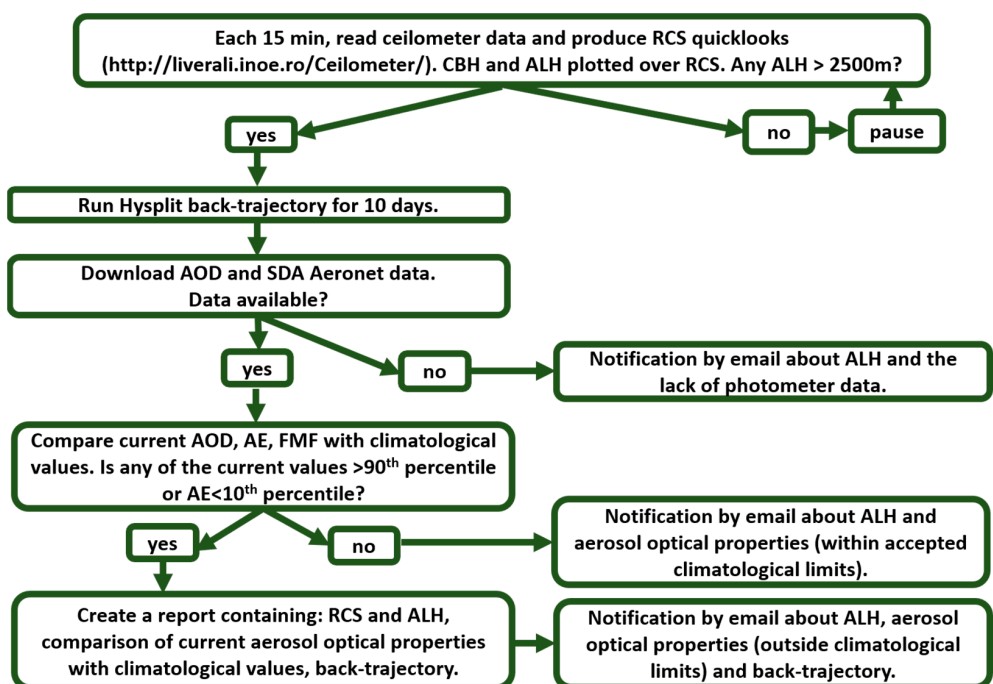

**Figure 1.** Methodology flowchart. RCS—range-corrected signal. CBH—cloud base height. ALH—aerosol layer height. AOD—aerosol optical depth. SDA—spectral deconvolution algorithm. AE—Angstrom exponent. FMF—fine-mode fraction.

If we find any ALH above 2500 m a.g.l. over the last 15 min, the HYSPLIT back-trajectory is computed. The input altitude (a.s.l.) for the back-trajectory is the mean of all second ALHs found over the last 15 min and the mean of all third ALHs when present. Note that we perform individual back-trajectory analysis for the second ALH and for the third ALH (when both are present). Next, we proceed to download the AOD and SDA files from AERONET website. The data downloaded cover the current and the previous day. We download the files containing level 1.0 and level 1.5, considering that level 2.0 is not available in NRT. The level 1.5 data are cloud-screened and quality controlled [70]. If no data are available, a message is sent by email, warning about the presence of the ALH while no AERONET data are available. This information is also saved in a log file. A description of the log file is given at a later stage.

If AERONET data are available, the closest photometer measurement over the last 3 h from the time of layer detection is considered. The variables described in Section 2.2. are analysed. The current values of AOD at 340 nm, 500 nm, 870 nm and 1020 nm, AE440/870, FMF, FMAOD and CMAOD at 550 nm are compared with the climatological values at our station [67]. This climatology was constructed using all the available data for each month over the ten years period. The comparison with the climatological values is performed as following. If any of the variables exceed 90th percentile or AE is bellow 10th percentile, the aerosol layer is assumed to be outside the climatological limits while it is labelled as high pollution and it will be closely monitored. If neither of the values are above 90th percentile nor the AE is below 10th percentile, an email is sent to notify about the presence of the ALH with the mention that the values are within the climatological limits. We chose to use percentiles over mean and standard deviation in order to have a better representation of our sample, considering that AOD and AE values do not follow normal distributions. The 90th percentile for the characterization of extreme AOD was selected so as to be consistent with other studies (e.g., [77–79]). The criteria defining the values of the current variables 'outside the climatological limits' are summarized in Table 1. If at least one of them occurs, the warning is issued stating that the current measurement is "outside climatological limits".

**Table 1.** Criteria to define the values outside the climatological limits.

| Variable | Wavelength [nm] |
|---|---|
| AOD > 90th percentile | 340, 500, 870, 1020 |
| AE < 10th percentile or AE > 90th percentile | 340/870 |
| FMAOD > 90th percentile | 500 |
| CMAOD > 90th percentile | |
| FMF > 90th percentile | 500 |

A report is created where the following information is included: a plot of AOD values (current and climatological values), a plot of AE, a plot of FMAOD and CMAOD, a plot of FMF (e.g., Figure 2), a slide containing the numerical values for the variables exceeding the climatological values, a figure containing RCS and CBH (e.g., Figure 3 upper plot), RCS and ALH (e.g., Figure 3, lower plot) and a plot with the back-trajectory as provided by HYSPLIT (e.g., Figure 4).

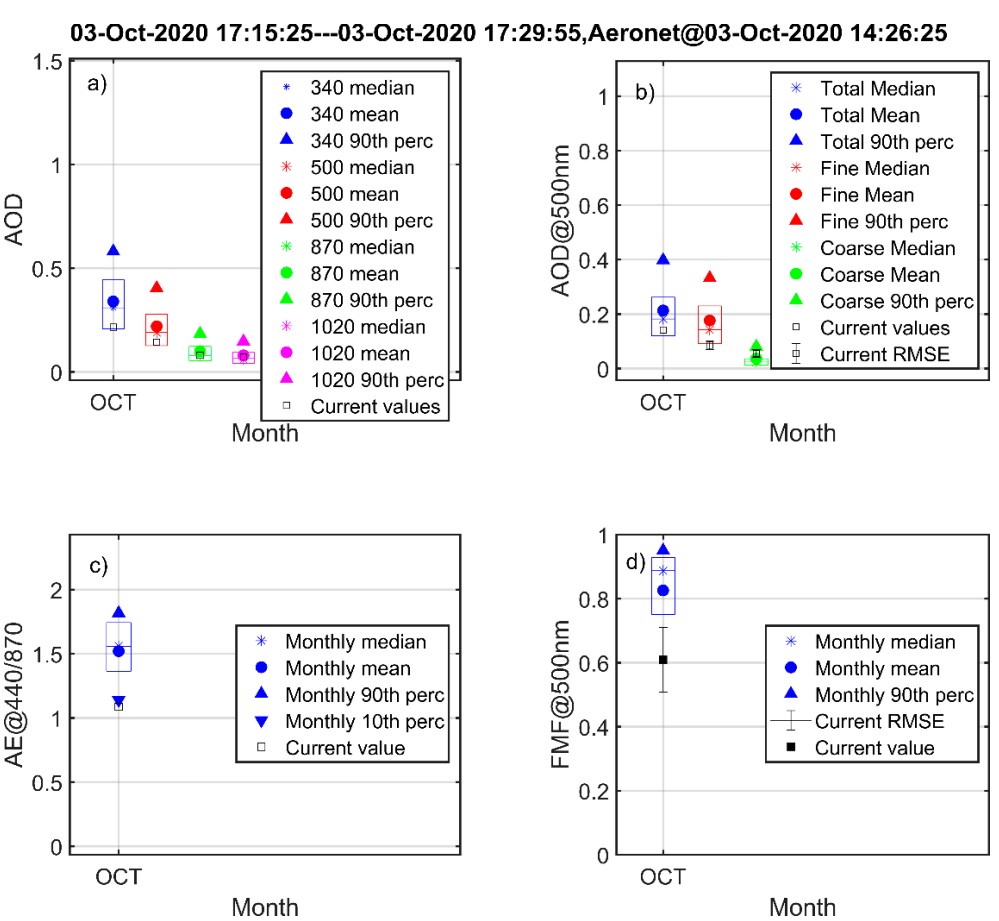

**Figure 2.** (**a**) AOD at 340 nm, 500 nm, 870 nm and 1020 nm. (**b**) Total AOD, FMAOD and CMAOD at 500 nm. (**c**) AE at 440/870. (**d**) FMF at 500 nm. The title shows the time interval over which ALH was detected and the time when AERONET data were available ('Aeronet@').

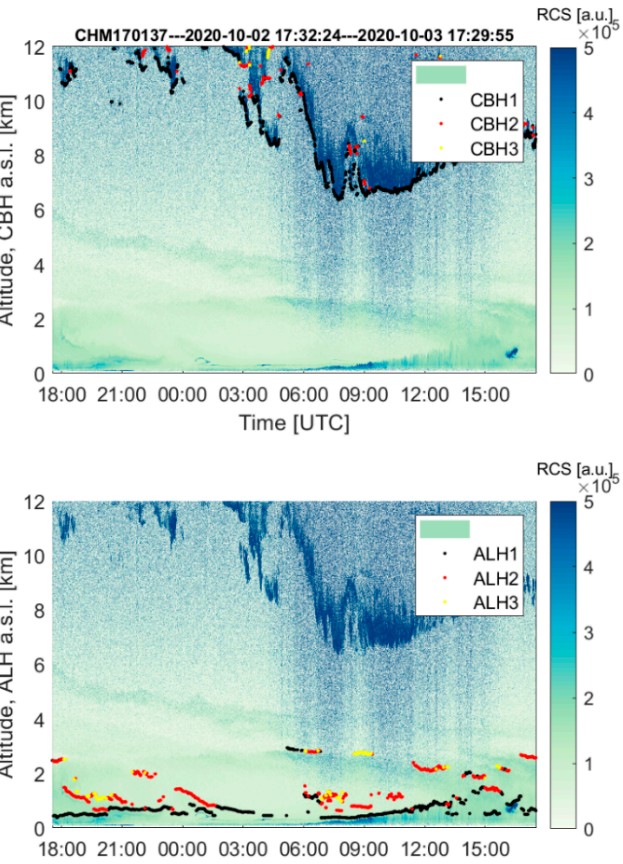

**Figure 3.** RCS over the last 24 h with overlaps of CBH (upper plot) and ALH (lower plot). The time interval for the last 24 h is shown in the title.

As limitations of the current version of the methodology we mention the following:

- The early detection warning system works solely when photometer data are available (no precipitation and at least some cloud-free periods). In cases when cirrus clouds are present and the cloud screening data are not issued over the last 3 h, we do not perform the analysis.
- The AERONET data are released the earliest in about 1 h from the date of recording (timeliness). Consequently, the photometer data are not quite NRT. The photometer data used in the study covers 3 h interval from the time of the layer detection.
- ALH detected are limited up to ~4–5 km. As mentioned, the altitude of the pollution layers is taken as the top of the pollution. In a new version, once we implement our algorithm for layers identification, we will consider the middle of the layer.
- The fixed threshold of the 2.5 km, in some cases is above the PBLH. This could result to missing some layers between the PBLH and 2.5 km. This issue is further discussed in Section 3.4, where we try to estimate the impact of the fixed height threshold at the number of possible layers that have not been identified by the algorithm.

The current alerting system is in place since 11 May 2018. Over this period, we found the most common unfavorable situations which occur. Thus, we fail to obtain the following information about lofted aerosol layers:

- AERONET data are not available. Most of the cases occur because the weather was not favorable. However, sporadically we encountered also technical issues such as a failure in the transmission of the data from photometer towards Photons (Lille, France).
- The HYSPLIT back-trajectory could not be performed. In most cases, error messages indicate the lack of the meteorological fields.

- Local issues with internet disruptions.
- In rare cases, electrical power cut-offs that affect the operation of the ceilometer.

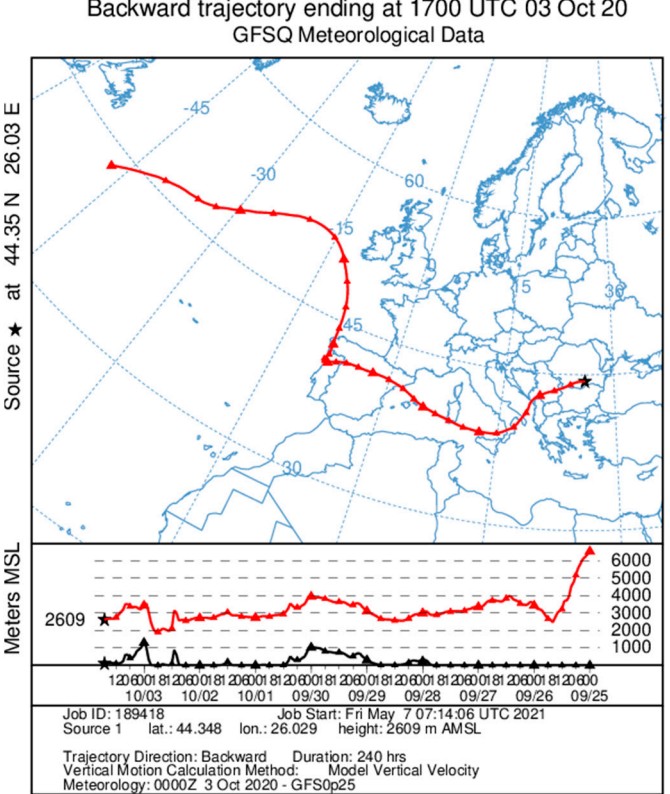

**Figure 4.** HYSPLIT back-trajectory for ALH = 2609 m a.s.l. at ending time 17:00 UTC on 3 October 2020. The black curve represents the terrain altitude a.s.l. along the air mass trajectory.

## 3. Results and Discussions

Appendix A shows example of notifications about the presence of ALH. The messages shown as example in Appendix A are collected in a text file while they are sent by email to designated scientists. Over the period of 20180511–20210511, a total number of 5167 messages were recorded.

### 3.1. Example Report

In this section we show an example of the plots and the information recorded by the automatic process. Figure 2a presents the AOD at four wavelengths, from UV to NIR: 340 nm, 500 nm, 870 nm and 1020 nm. The current values are shown by the black open squares, while the climatological values are shown in color code (see legend). For climatological values, the median, the mean and the 90th percentiles are illustrated along with the box plots showing the 25th and 75th percentiles. Figure 2c presents the current AE values (black square) along with the respective climatological values. For this example, we observe that the current AE value is below the 10th percentile. Low values of AE indicate high load of coarse particles. In turn, the presence of the coarse particles can help identify the aerosol type, since coarse particles, for example, are characteristic of dust, while fine particles are characteristic of smoke or urban pollution. Directly related with AE values, the CMAOD at 500 nm shows high value, above the 90th percentile (Figure 2b) while FMF at 500 m represents 60% of the load (i.e., 40% coarse-mode fraction) (Figure 2d). Note that RMSE (root mean square error) provided by the SDA files, represents the estimated error (derived from estimated AOD accuracy). All the values outside the climatological values seen on Figure 2 are recorded in the report. In this example, AE440/870 is slightly smaller

than the 10th percentile, i.e., AE440/870 = 1.0517 < AE440/870p10 = 1.1395 (Figure 2c). For our site, the predominant type is fine-mode aerosols [73], thus cases with high load of coarse particles are indicative of long-range transported pollution. Figure 3 shows RCS along with CBH and ALH superimposed (upper and lower plot, respectively). In the lower plot, we see the marks of the second ALH over the last 15 min (red dots). Figure 4 shows the HYSPLIT back-trajectory for ALH = 2609 m a.s.l. Please note that due to a re-rerun of the back-trajectories, the 'Job Start' does not correspond to the NRT measurement.

### 3.2. Some Statistics over the Colleted Data

Over the three-year period (from 11 May 2018 to 11 May 2021), 5167 events were recorded. In this study, an event corresponds to a specific detection of an ALH from a single measurement (over 15 min). Thus, the current definition differs from the commonly usage which may mean the presence of aerosol load in the free troposphere from a specific source for a specific time interval. From these, the following statuses were recorded. For 4036 events of ALH, no AERONET data were available. From the remaining 1131 notifications with AERONET data, 556 measurements were labeled as 'within climatological limits' while the rest of 575 were labeled 'outside climatological limits'. From the later, for a number of 32 measurements, the report couldn't be created due to the technical difficulties described above. The 575 events labelled as high pollution occurred during 84 days over the three years monitoring period. Considering an episode made of a series of events which are not connected with neighbouring events (i.e., the distance between two episodes is >1 day), we obtain 45 pollution episodes. Comparing the daily averages of the optical parameters with the climatological values we found 152 days outside the climatological limits. 38 days of those were part of the 84 highly polluted days we found through alerts. Thus, for 46 days, the high pollution was based on instantaneous values at certain times (while the daily average was inside climatology). On the other hand, the remaining 114 days revealed through the daily averages analyses can be related with the contribution of the PBL to the atmospheric column.

A number of 5167 back-trajectories were calculated over the period May 2018 to May 2021 (Appendix B, Figure A1). The 10 days back-trajectories cover Europe but also N. Africa, W. Asia and to a less extend N. America. Figure 5 shows the histogram of the measurement altitude (a.g.l.). The most common altitude of the pollution layer is between 2500 and 2600 m a.g.l. Recall that the Lufft algorithm provide ALH up to ~4–5 km.

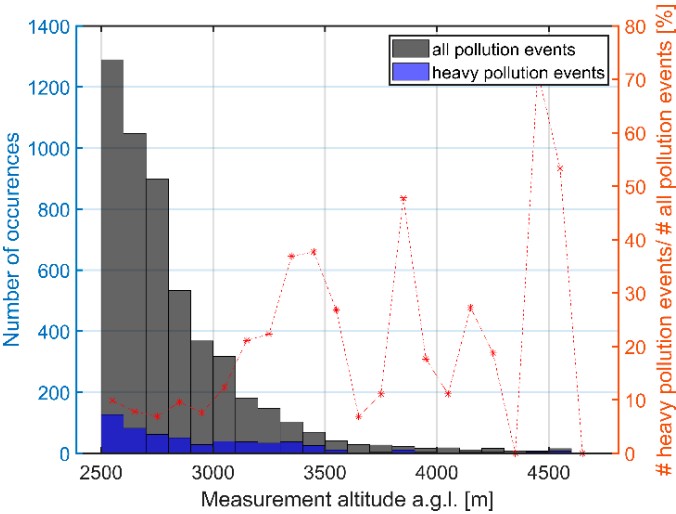

**Figure 5.** Histogram of the measurement altitude. The right axis shows the percentage of the heavy pollution events with regard to the total number of events for each bin size.

The histogram for the total number of altitudes (black) decreases exponentially towards higher altitudes while the number of heavy polluted events (outside the climatologi-

cal limits) has a different trend. The percentage of the heavy polluted layers with respect to the total number of layers in a specific bin size is shown on the right axis (red). 80% of the total number of layers resides below 3 km altitude. For the heavy polluted layers, 60% of them resides below 3 km altitude. In a recent study, Radenz et al. [80] evaluated the air mass source attribution based on normalized residence time of the air mass. The normalized residence time represents the time the airmass passes over a region below 2 km a.g.l. with regard to the total travel time (here, 240 h). Radenz et al. describes the method and shows examples for both HYSPLIT and FLEXPART models. The method can be applied for either a single trajectory or an ensemble. Thus, the authors claim to assess an estimate of the air mass source. It is assumed that the possible surface effects on the air parcel occur when the air mass descends below 2 km a.g.l. The sources are either assessed in light of geographical regions (Europe, Sahara, Arabian Peninsula, etc.) or land cover following MODIS classification (e.g., water, forest, urban, savanna, etc.).

Following Radenz et al. 2021 [80], we assessed the normalized residence time over various continents and further over regions with the same type of aerosol emissions. Considering the limited available information, the aerosol type is simply divided in three main categories based on the source region types: marine (with source regions the sea and/or the ocean), dust (with the source region the deserts), and continental (with the source regions the continents, except the desert areas). Since we do not have additional information to make a proper assignment of the aerosol type, the following numbers are mostly qualitative and not quantitative. The Biomass burning is included in the continental type, while the smoke can originate either from Europe, Asia, or US (e.g., [23]). The dust contribution to the aerosol type is due to the air mass passing over Africa and the Arabian Peninsula. For the latter, we considered sources from the following areas: Saudi Arabia, AUE, Kuwait, and Iraq. The source was considered marine if the air parcels travel over ocean (Atlantic) or seas (mostly Mediterranean and Black Sea). The continental sources cover mostly Europe but also America and Asia (except those countries considered for dust).

Figures 6 and 7 show the monthly averages for the normalized residence time over continents (a) and according to the aerosol type (b) for all the events (Figure 6) and for the events labelled 'outside climatology limits' (Figure 7). For all the events, the predominant aerosol type is continental, with a major contribution from Europe. The overall average residence time is 31%, 7%, 2%, 10%, and 50% over water (sea and/or ocean), Africa, America, Asia, and Europe, respectively. This corresponds to a 31%, 9%, and 60% contribution from the marine, dust, and continental types, respectively.

We then assessed the heavy pollution ('outside climatology limits') over 575 events. The contribution from various continents is 15%, 6%, 2%, 22%, and 55%, which correspond to water, Africa, America, Asia, and Europe, respectively. The aerosol-type contribution is 15%, 12%, and 73% for the marine, dust, and continental types, respectively. Thus, the continental aerosol represents a major contribution to the heavy pollution.

The plots showing the individual events are shown in Appendix C (Figures A2 and A3).

According to Georgoulias et al. [81], the contribution to AOD for Northern Balkans (including Bucharest and Eforie in Romania) consists of 59% anthropogenic aerosol, 23% dust, and 18% fine-mode aerosol (natural sources). According to Carstea et al. [73], the fine mode is dominant for the Bucharest–Magurele site, whereas the authors attribute it to anthropogenic origin. The fine mode is characteristic of the biomass burning and anthropogenic pollution, such as traffic, residential heating, industrial emissions, and agriculture. [82,83]. Thus, broadly speaking, the aerosol over the region is of continental origin.

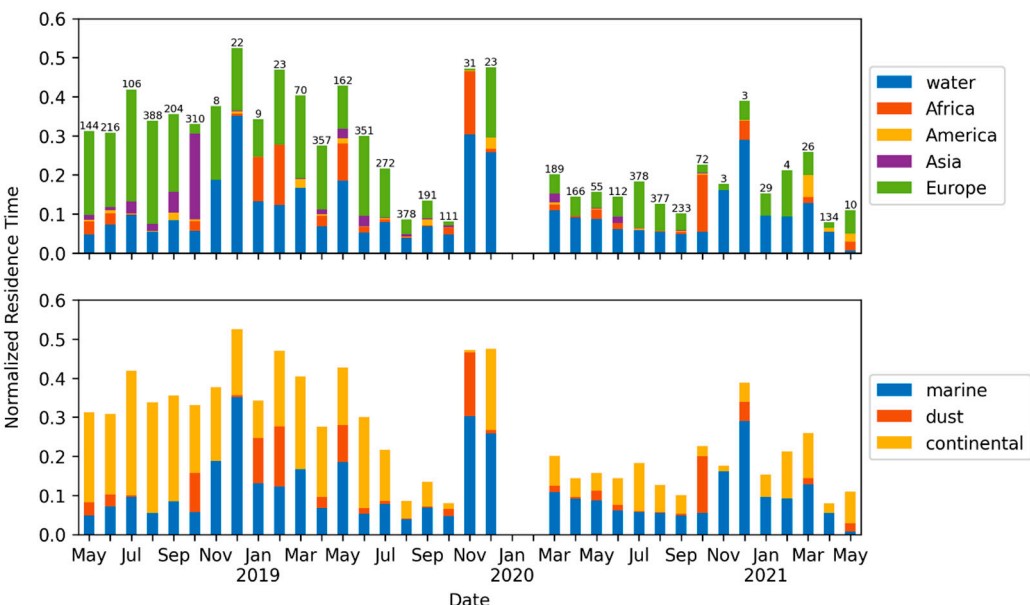

**Figure 6.** Normalized residence time over continents (**upper plot**) and the associated aerosol type (**lower plot**). The numbers on top of the bars represent the number of cases for each month. Water stands for sea and/or ocean.

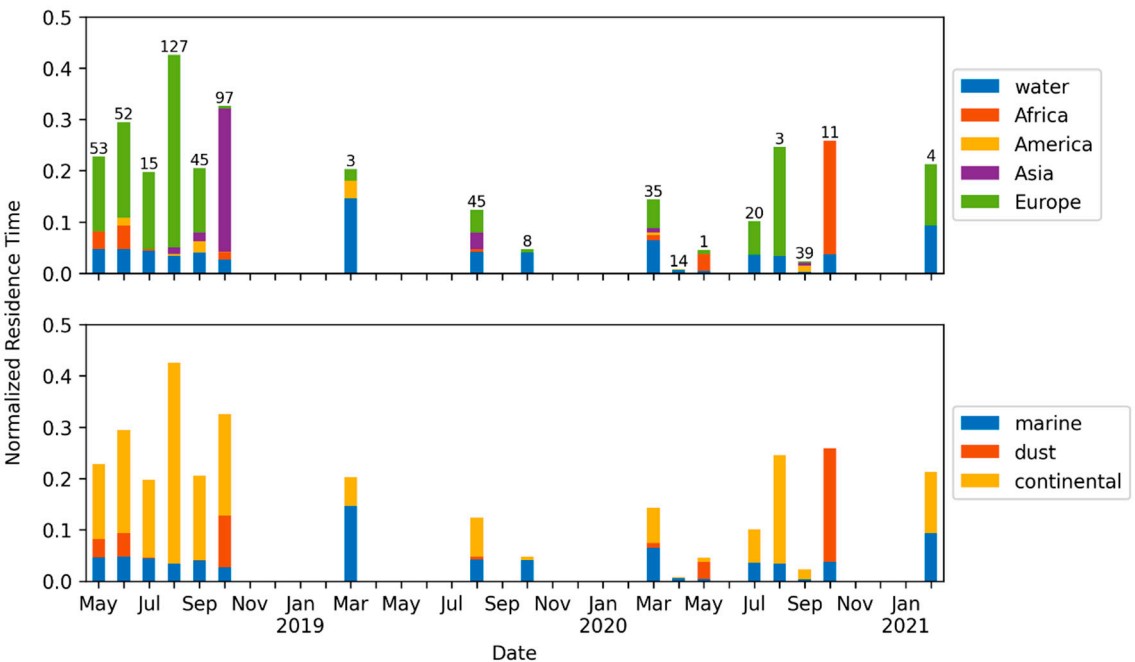

**Figure 7.** Same as Figure 6 for the heavy pollution events ('outside climatology limits').

The statistics over the seasons show the following: 113 events recorded over winter (DJF), 1313 events recorded over spring (MAM), 2578 events recorded over summer (JJA), and 1163 events recorded over autumn (SON). The statistics by number of events recorded in each month are given in Figure 8. The highest number of recorded events is found in August, followed by July, June, April, September, October, May, and March. The smallest number of events is observed in February. If we represent these numbers normalized by the total number of 15 min intervals over three-year measurements, we obtain the percentage of the events recorded (Figure 8b). In the majority of months, the recorded events represent less than 1.5% of the total number of 15 min intervals. For all months, this represents 10.6% of the total number of 15 min measurements. In particular, with regards to the total number

of 15 min intervals in a specific month, ~13% of events were in August, while ~0.3% of events were in February.

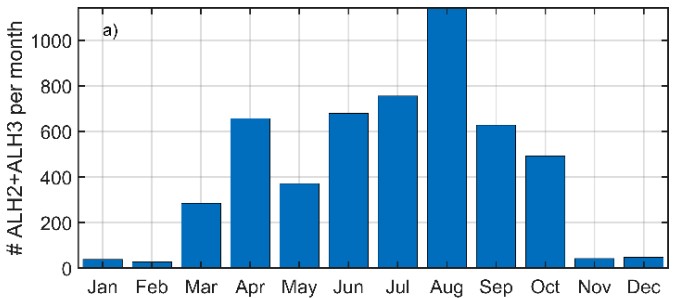

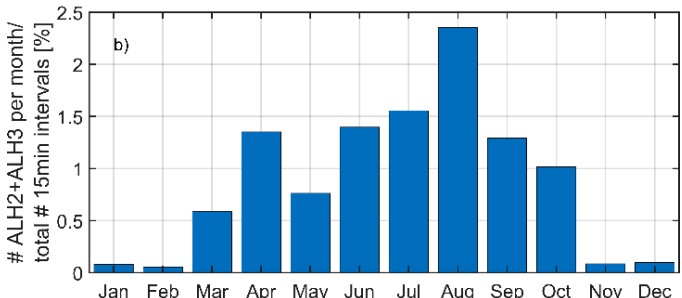

**Figure 8.** (**a**) Number of events (ALH2 + ALH3) recorded for each month. (**b**) Number of events per month normalized by the total number of 15 min intervals over the three-year period.

A representation of the total number of cloud-free intervals over the three years (Figure 9) shows that, in the large majority of the cases, ALH2 and ALH3 lay below 2500 m, especially during winter months. This is a clear indication that the presence of the low-level clouds or fog during the winter time is not the main reason why a small number of events were recorded; however, the cloud coverage over Magurele is indeed higher during these months [84] and an increased number of fog events is observed [85]. However, during winter, the PBLH can be much lower than the threshold of 2500 m; thus, many layers above PBL are missed. In Section 3.4, we discuss the ALH occurrences below 2500 m with respect to the PBLH.

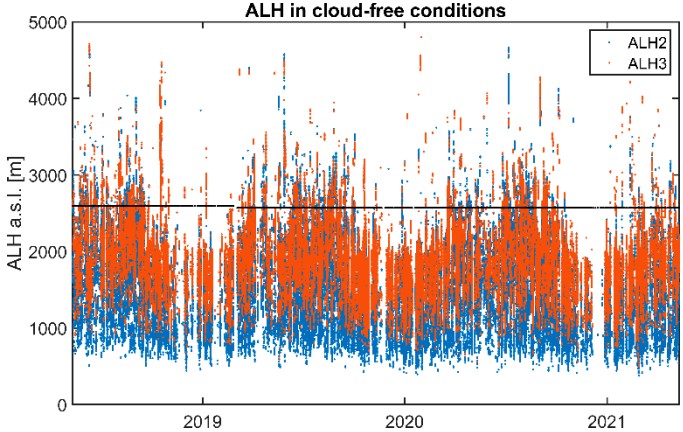

**Figure 9.** Number of ALH2 and ALH3 in cloud-free conditions over the entire three-year period. The black horizontal line shows an altitude of 2500 m a.g.l., which is the lower limit for our algorithm.

### 3.3. Case Study

Figure 10 shows RCS for 11 June 2018. During this day, five warnings were issued. The warnings were sent between 05:39 and 06:14, while the ALH2 or ALH3 were around

4200 m. For all the events, the AE440/870 was < AE440/870p10, while FMF shows values around 50%, suggesting a large contribution from coarse particles. Note that the layer was first observed on 8 June, while warnings were issued from 10 June. However, the first lidar measurements were obtained on 11 June. MERRA-2 reanalysis [86] shows the presence of dust over our site for the 8–11 June 2018 period (Figure 11). The AERONET SDA fine- and coarse-mode AOD (not shown here) shows similar values for both the fine mode and the coarse mode.

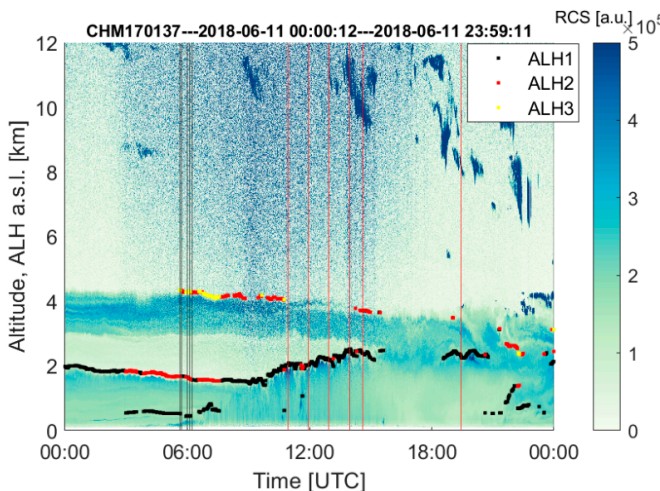

**Figure 10.** RCS for 11 June 2018 with overlaps of ALH. The vertical black lines delimitate the middle of 15 min intervals where warnings were automatically issued from our algorithm, while the vertical red line shows the middle of a ~1 h measurement interval by lidar.

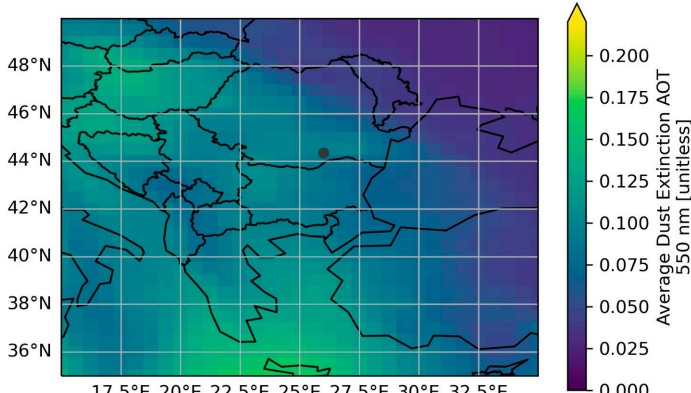

**Figure 11.** Average dust extinction AOT over the 8–11 June 2018 period provided by MERRA-2 reanalysis (source: https://giovanni.gsfc.nasa.gov/giovanni/ (accessed on 9 March 2021)). The dot shows the location of our ground-based station.

The back-trajectories show air masses passing over North Africa with a higher residence time (see the values below) for the trajectories corresponding to the third, fourth, and fifth warnings. Figure 12 shows two back-trajectories, corresponding to the starting time at 05:00 and 06:00 UTC for a layer at altitude of ~4200 m. The air mass residence time, based on Radenz's procedure for the five back-trajectories, is: (I) 0.36, 0.16, and 0.01; (II) 0.29 0.17, and 0.01; (III) 0.1, 0.37, and 0.004; (IV) 0.01, 0.30, and 0; and (V) 0.05, 0.49, and 0 for the marine, dust, and continental aerosol types, respectively. Thus, based on the origin of the airmasses, we observe a mixture of dust and marine (while the continental contribution is much smaller) with a higher dust contribution for the last three cases. When lidar measurements of depolarization and lidar ratio are available, one can estimate the contribution of marine and dust (see [87], Table 3). Figures 13 and 14 show the lidar measurements taken

during 11 June 2018, over the 11:26–12:27 UTC and 19:00–19:30 UTC periods. During day time, the Raman measurements are not available. For the measurements during day, we evaluated the following mean values for the 3000–4000 m layer (2500–3500 m for night time), as shown in Table 2: the backscatter Angstrom exponent (BAE) between 355 nm and 532, the backscatter Angstrom exponent between 532 nm and 1064 nm, and the particle linear depolarization ratio (PDR). When Raman measurements are available (night-time), we calculated in addition the extinction Angstrom exponent (EAE) and the lidar ratios (LRs) at 355 nm and 532 nm. According to Papayannis et al. [88], which studied the Saharan dust through EARLIENT measurements, BAE values for the ratio 532/1064 are found between −0.5 and 3, while PDR values range from 10% to 25%. LRs at 532 nm are found between 20 and 100 sr. Gross et al. [87] found depolarization values as high as 32 ± 2%. Typical dust values for EAE (literature research, measured), shown in a study by Nicolae et al. ([89], Table 6), range between −0.15 and 2.5.

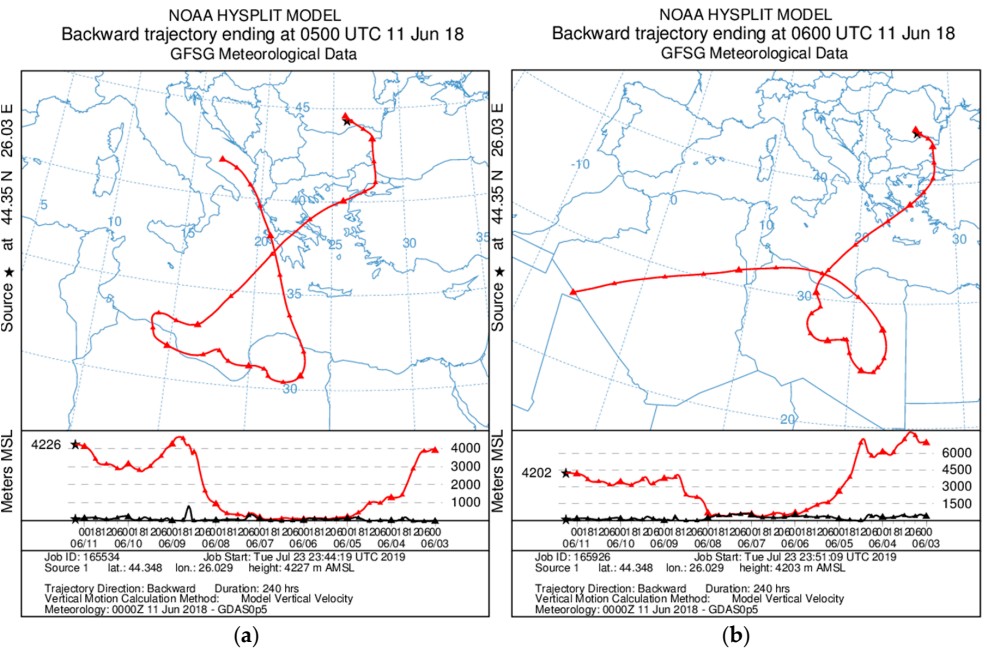

**Figure 12.** Air mass back-trajectory for layers at ~4200 m, with staring times at 05:00 UTC (**a**) and 06:00 UTC (**b**).

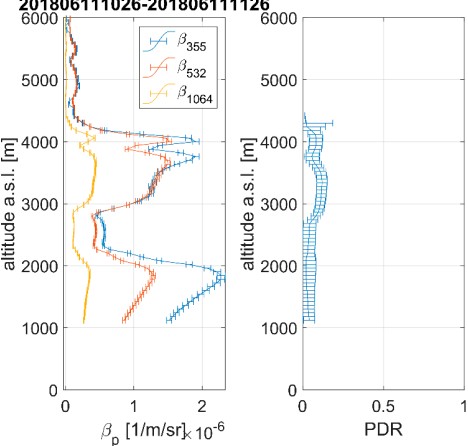

**Figure 13.** Lidar measurements around 12 UTC: particles backscatter coefficient for 355, 532, and 1064 nm (**left**) and particle linear depolarization ratio at 532 nm (**right**).

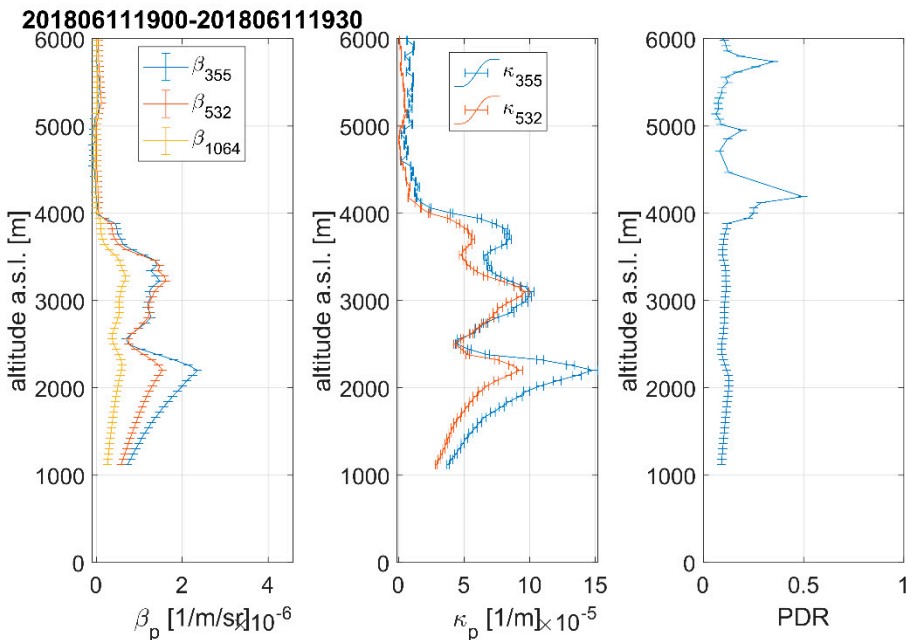

**Figure 14.** Lidar measurements around 19:15 UTC: particle backscatter coefficient for 355, 532, and 1064 nm (**left**); particle extinction coefficient for 355 and 532 nm (**middle**); and particle linear depolarization ratio at 532 nm (**right**).

**Table 2.** Mean optical properties and their standard deviation retrieved from lidar measurements over 3000–4000 m (2500–3500 m for the night measurement).

| Optical Property/ Measurement Time | PDR [%] | BAE 355/532 | BAE 532/1064 | LR355 [sr] | LR532 [sr] | EAE |
|---|---|---|---|---|---|---|
| 10:26–11:26 | 14 ± 2 | 0.1 ± 0.41 | 0.95 ± 0.16 | | | |
| 11:26–12:27 | 12 ± 2.3 | 0.29 ± 0.34 | 1.78 ± 0.19 | | | |
| 12:27–13:27 | 15 ± 3.5 | 0.15 ± 0.42 | 1.67 ± 0.19 | | | |
| 13:27–14:28 | 15.5 ± 3 | 0.03 ± 0.39 | 1.83 ± 0.22 | | | |
| 14:28–14:47 | 18 ± 3.6 | 0.01 ± 0.3 | 1.9 ± 0.33 | | | |
| 19:01–19:30 | 10.6 ± 0.6 | -0.04 ± 0.20 | 1.24 ± 0.06 | 65 ± 12 | 57 ± 12 | 0.34 ± 0.28 |

Here, we demonstrate the usefulness of our early detection algorithm at identifying and monitoring a Saharan dust event over Magurele. The first layer was observed on 8 June and reports with values outside the climatological limits were first issued on 10 June. Lidar measurements performed on 11 June gave typical values of dust for the optical parameters at the identified layer, while the residence time of the back-trajectories performed for altitudes observed in ceilometer was higher over North Africa for the last three altitudes. In addition, assimilation data from the MERRA-2 confirmed the presence of dust over our site.

### 3.4. Lessons Learnt over the Three-Year Testing Period

In this paragraph, we evaluate the developed algorithm based on the experience acquired during the first three years of its testing period and we discuss future upgrades to improve the system.

The very first notice is that the number of alerts is excessive for persistent pollution events (e.g., lasting few days). This could be resolved by allowing the system to have an option to set the minimum interval between alerts (from the predefined 15 min interval) so as to adapt to the requirements of each site. This change would also allow the system to perform more advanced model runs, e.g., to calculate HYSPLIT ensemble back-trajectories instead of a single trajectory, which require more time to finish. As of July 2021, we switched to 30 min alert intervals.

Another issue that could affect the quality of the current version of the system is the fact that we used a predefined threshold of 2500 m for the free troposphere layers. The choice of this altitude was based on the fact that, generally, PBLH is below this altitude at our site and ALH1 is the top of PBLH in most cases. However, this could impact in missing some layers below 2500 m when the PBLH is significantly lower, especially during the winter time. To assess the impact of the predefined altitude, an off-line process to calculate the PBLH (which includes the residual layer) was run for all our events. Figure 15 shows the retrieval of the PBLH following the methodology developed by Wang et al. [90]. PBLH was determined as an average over 30 min (48 values daily). ALH2 and ALH3 used in this analysis are also shown. Thus, in over 5150 common cases (over 431 days), i.e., 12.2% of cases, ALH2 and ALH3 lied below PBLH. If we consider all 30 min intervals for each of the 431 days (when we had notifications) (not shown here), we found that ALH2 (ALH3) lied below 2500 m in 65% of (36%) cases. Comparing with PBLH, we found that 24% (10%) of ALH2 (ALH3) is below PBLH. The percentages for ALH2 (ALH3) lying between PBLH and 2500 m are 42% (27%).

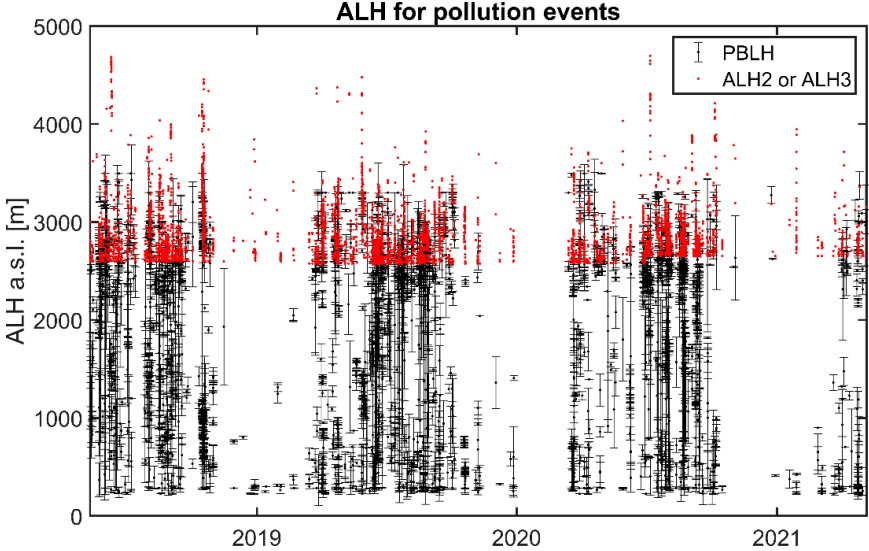

**Figure 15.** PBL height retrieved with in-house algorithm. ALH2 and ALH3 provided by a ceilometer are shown as well. The error bar for PBL height represents the std ($\pm 1\ \sigma$) over a 30 min average.

An example of the missing ALH (2 or 3) is visible in Figure 10 where the current algorithm does not catch the ALH around 4 km from 00:00 to ~06:00 UTC.

Based on current findings, the following procedures will be implemented in order to improve the current automatic algorithm:

- retrieval of the PBLH in NRT which allows the search of ALH above it, thus eliminating the constraint of 2500 m. Once the PBLH is estimated, the number of cases where ALH found in PBL will be eliminated;
- retrieval of the ALH following the in-house developed algorithm allows to search for ALH above PBLH within FT, depending on the SNR of the ceilometer;
- determination of the layers' air mass source, following Radenz et al. [80];
- update of the aerosol climatology of the optical properties derived from the photometer;
- assessment of the contribution of the PBL and free troposphere in the total AOD.

## 4. Conclusions

The current early detection system is implemented at INOE 2000 in Magurele (Romania). It is based on continuous measurements taken by a CHM15k ceilometer and a CE318 photometer. Climatological values (based on ten years of measurements) of the monthly means of the aerosol optical properties revealed by the photometer are used to determine

the degree of the current pollution. Thus, alerts are sent each time an ALH is seen above 2500 m a.g.l. When photometer data are available, the alert messages contain the warning message 'outside climatological values' when one of the optical properties are above the 90th percentile of the monthly climatological values or when the value of AE is also below 10th percentile. HYSPLIT back-trajectory is performed to determine the pollution sources. Over three years of measurements, 5167 notifications were recorded. For 4036 events, no AERONET data were available. For the other 1131 events with AERONET data, 556 events were labeled as 'within climatological limits', while the other 575 were labeled as 'outside climatological limits'.

Following Radenz [80], we assessed the off-line aerosol type based on the residence time of the air mass. For all the events, the contribution to the aerosol pollution was 31%, 9%, and 60% from the marine, dust, and continental types, respectively. For the events labeled 'outside climatology limits', the contribution was 15%, 12%, and 73% for the marine, dust, and continental types, respectively. Thus, the continental aerosol dominates the heavy pollution.

One of the main limitations of the current approach is the fact that the layers lying between PBL and 2500 m are missed while the layers above ~4000–5000 m are not caught by the manufacturer's algorithm. Furthermore, the AERONET data are not available during unfavorable meteorological conditions and during night time periods when the lunar photometer does not provide measurements.

Further considerations related with the improvement in the present methodology are related to the implementation of our in-house algorithm to determine the PBLH and the ALH in the whole troposphere. In a second stage of the pollution warnings, when the pollution event is detected and classified as an extreme event, we will consider research instruments to further analyze the degree of pollution. Therefore, a multi-wavelength Raman lidar will acquire measurements and more optical properties will be determined, such as aerosol backscatter or extinction coefficients at two/three wavelengths and particle depolarization ratios. Intensive parameters, such as lidar ratio and extinction Angstrom exponents along with particle linear depolarization ratio, help to identify the aerosol types (e.g., [90]).

**Author Contributions:** Conceptualization, M.A. and K.F.; methodology, M.A.; software, M.A. (all data analysis), I.B. (HYSPLIT runs), I.S.S. and D.W. (PBLH retrieval), L.B. and V.N. (lidar data processing); validation, M.A.; formal analysis, M.A. (ceilometer), D.W. (PBLH), L.B. (lidar); investigation, K.F., I.B., L.B. and I.S.S.; resources, M.A., K.F., I.B., and I.S.S.; data curation, M.A. and K.F.; writing—original draft preparation, M.A.; writing—review and editing, K.F., I.B. and I.S.S.; visualization, M.A.; supervision, K.F.; project administration, M.A.; funding acquisition, M.A. All authors have read and agreed to the published version of the manuscript.

**Funding:** This work was supported by the Romanian National contracts 18N/08.02.2019 and 18PFE/30.12.2021, and partly by European Regional Development Fund through Competitiveness Operational Programme 2014–2020, Action 1.1.3 creating synergies with the H2020 Programme, project H2020 Support Centre for European project management and European promotion, MYSMIS code 107874, ctr. no. 253/2.06.2020 and European Regional Development Fund through Competitiveness Operational Programme 2014–2020, POC-A.1-A.1.1.1-F-2015, project Research Centre for Environment and Earth Observation CEO-Terra, SMIS code 108109 and The Executive Agency for Higher Education, Research, Development and Innovation Funding (UEFISCDI), PN-III-P2-2.1-PED-2019-1816. The development of the PBLH codes was funded within the European Space Agency (ESA-ESTEC), grant no. 4000119961/16/NL/FF/mg (POLIMOS).

**Acknowledgments:** We thank AERONET (PHOTONS) for instrument calibration and maintenance of the Cimel instrument and AERONET (GSFC) for processing and disseminating these data and NOAA Air Resources Laboratory (ARL) for the provision of the HYSPLIT transport and dispersion model and/or READY website (http://www.ready.noaa.gov, last access: 12 July 2021) used in this publication. We also acknowledge Alexandru Dandocsi, Anca Nemuc, Sorin Parloaga, and Razvan Parloaga for support with photometer and ceilometer administration and technical support.

**Conflicts of Interest:** The authors declare no conflict of interest.

**Abbreviations**

List of acronyms:

| | |
|---|---|
| ACTRIS | Aerosol, Clouds and Trace Gases Research Infrastructure |
| AD-NET | Asian Dust and Aerosol Lidar Observation Network |
| AE | Ångström exponent |
| AERONET | Aerosol robotic network |
| a.g.l. | above ground level |
| ALH | Aerosol layer height |
| AOD: | Aerosol optical depth |
| a.s.l. | Above sea level |
| BAE | Backscatter Angstrom exponent |
| CBH | Cloud base height |
| CloudNet | Cloud Network |
| CMAOD | Coarse-mode aerosol optical depth |
| EAE | Extinction Angstrom exponent |
| EARLINET | European Aerosol Research Lidar Network |
| EEA | European Environment Agency |
| EPA | Environmental Protection Agency |
| E-PROFILE | EUMETNET Profiling Programme |
| EUMETNET | European National Meteorological Services |
| FMF | Fine-mode fraction |
| FMAOD | Fine-mode aerosol optical depth |
| FT | Free troposphere |
| GDAS | Global Data Assimilation System |
| GFS | Global Forecast System |
| HYSPLIT | Hybrid Single-Particle Lagrangian Integrated Trajectory model |
| INOE 2000 | National Institute of Research and Development for Optoelectronics |
| LALINET | Latin America Lidar Network |
| LR | Lidar ratio |
| MARS | Magurele Centre for Atmosphere and Radiation Studies |
| MERRA-2 | Modern-Era Retrospective Analysis for Research and Applications, Version 2 |
| MODIS | Moderate resolution imaging spectroradiometer |
| MPLNET | NASA Micro-Pulse Lidar Network |
| NDACC | Network for the Detection of Atmospheric Composition Change |
| NIR | Near-infrared region |
| NRT | Near real time |
| PBL | Planetary boundary layer |
| PBLH | Planetary boundary layer height |
| PDR | Particle linear depolarization ratio |
| PM | Particulate matter |
| RADO | Romanian atmospheric 3D Research Observatory |
| RCS | Range-corrected signal |
| RMSE | Root mean square error |
| ROLINET | Romanian LIdar NETwork |
| SDA | Spectral Deconvolution Algorithm |
| VAAC | Volcanic Ash Advisory Centre |

**Appendix A**

*Appendix A.1. Notification about the Presence of ALH and the Absence of AERONET Data*

Below is an example of the message sent by email in situations when an ALH is detected but no AERONET data are available. The following information is provided: the time of detection, the type of layer (IInd or IIIrd), the value of ALH, and the interval over which it was observed.

> *20210128T065252 IIIrd ALH detected above 2.5 km a.g.l. (3.6462) over last 15 min, between 28-Jan-2021 06:47:19 and 28-Jan-2021 06:49:49UTC; no Aeronet data lev 1.5 in the last 3 h from 20210128T064719; keep an eye!*

*Appendix A.2. Notification about the Presence of ALH and the Values of the Optical Properties within the Climatological Limits*

The following example is for cases when AERONET data are available and the analysis over photometer variables show that their values are within the climatological limits. The time interval over which AERONET data are considered is mentioned as well as the AERONET file (e.g., 20210127_20210128_Magurele_Inoe).

*20210128T130207 IIIrd ALH detected above 2.5 km a.g.l. (3.876) over last 15 min, between 28-Jan-2021 12:58:50 and 28-Jan-2021 12:59:50UTC; Aeronet data lev 1.5 found in-20210127_20210128_Magurele_Inoe; within climatological limits; keep an eye!*

*Appendix A.3. Notification about the Presence of ALH and the Values of the Optical Properties outside the Climatological Limits*

The last example is for cases when AERONET data are available and the analysis over photometer variables show that at least one variable is outside the climatological limits. Information on the report file created is also given (location and name).

*20210227T140206 IInd ALH detected above 2.5 km a.g.l. (2.6658) over last 15 min, between 27-Feb-2021 13:54:01 and 27-Feb-2021 13:59:31UTC; Aeronet data lev 1.5 found in-20210226_20210227_Magurele_Inoe; outside climatological limits; see file "\\172.16.1.15\Workspace\Documents\03-Stiintific\Analize\Mariana\CHM15k \CHM170137\PollutionEvents\20210227T135401_20210227T135931_IIndALH.pptx"; analyse results and take action!*

## Appendix B

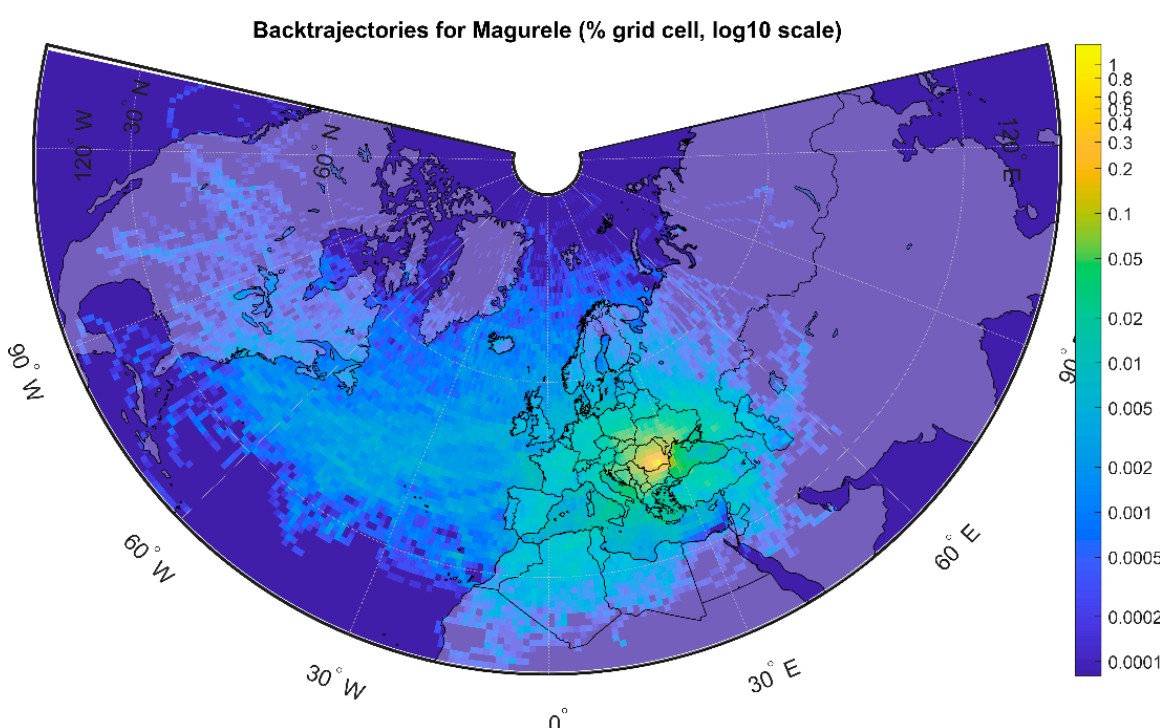

**Figure A1.** HYSPLIT back-trajectories for 5167 events. The % grid cell represents the number of observations in a grid cell to the total number of observations.

## Appendix C

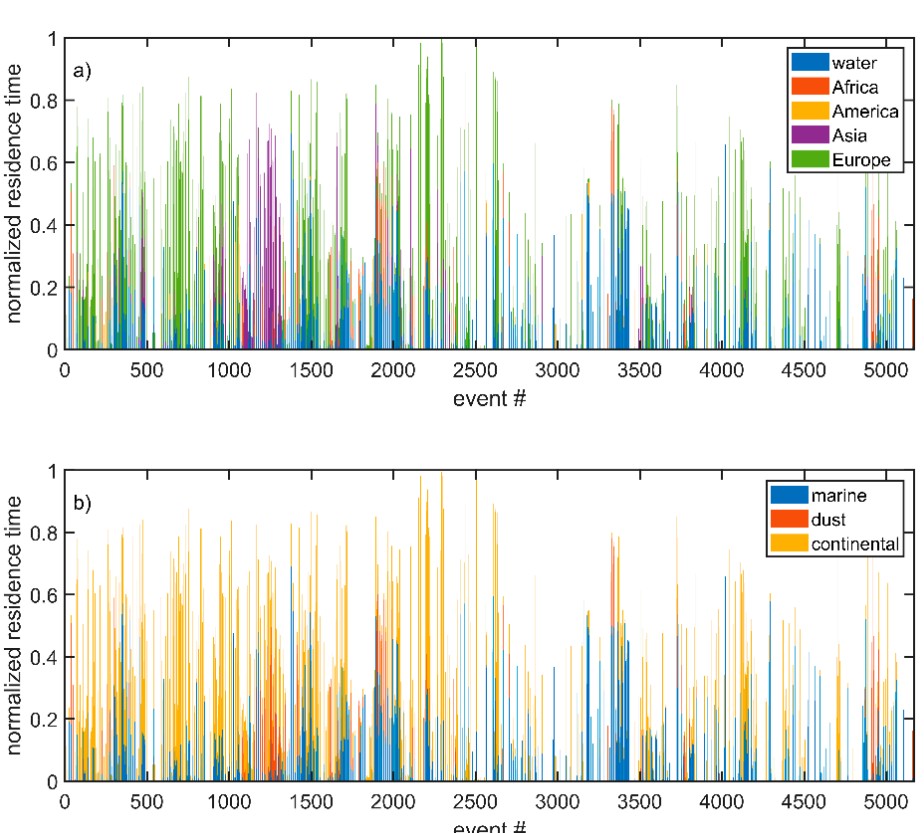

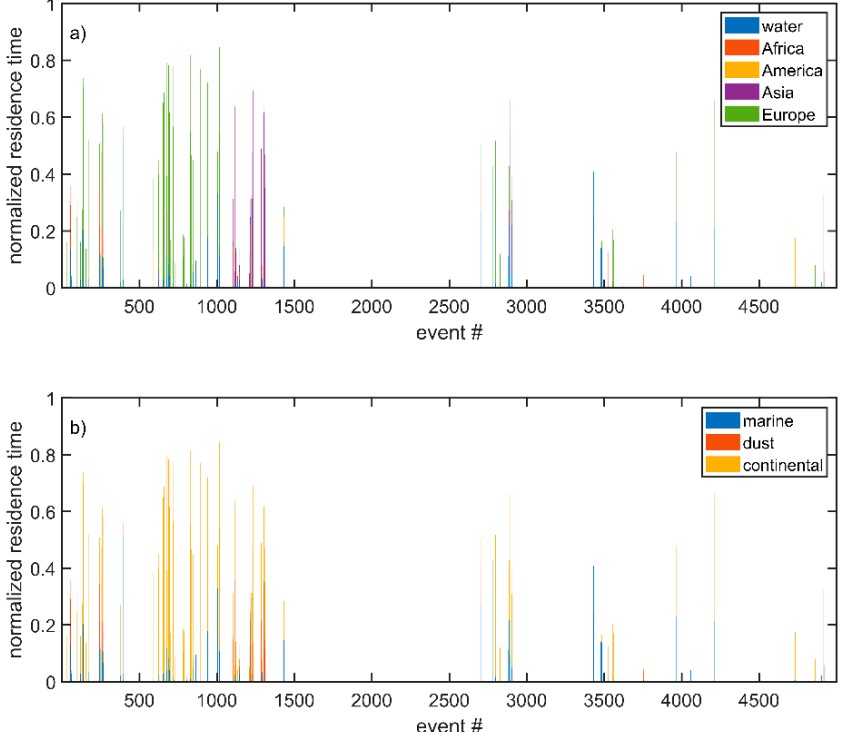

**Figure A2.** Normalized residence time over continents (**a**) and the associated aerosol type (**b**).

**Figure A3.** Same as Figure A2 for the heavy pollution events. Normalized residence time over continents (**a**) and the associated aerosol type (**b**) ('outside climatology limits').

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
