# Peer review of "Towards Early Detection of Tropospheric Aerosol Layers Using Monitoring with Ceilometer, Photometer, and Air Mass Trajectories"

_remotesensing, doi:10.3390/rs14051217_

Round 1

Reviewer 1 Report

Also based on the comments of the other three reviewers, authors didn’t address the suggestion to change the objective of the work presented in the paper and they made only minor revisions following the comments of the other three reviewers.  Even if I think that the frequent missing of radiometric data together with the usage of the level 1.5 data is a relevant flaw of the presented methodology, I decided to conform my decision to the decisions of the other three reviewers accepting the manuscript in its revised form. Indeed, some improvements were made by the authors thanks to the other reviewers comments and suggestions.

Author Response

We would like to thank Reviewer 1 for his previous review that helped us to improve our manuscript and for supporting its publication.

Reviewer 2 Report

I have no further comments on the manuscript

Author Response

We would like to thank Reviewer 2 for his previous review that helped us to improve our manuscript and for supporting its publication.

Reviewer 3 Report

The manuscript describes a combined system of aerosol early detection which is extended for the monitoring of the free troposphere aerosol distribution.  The problem solved in the manuscript is urgent and is of interest for the readers of Remote Sensing. 

I have only one minor comment: the abbreviation HYSPLIT (line 20) is not explained, whereas other abbreviations are all explained in the text of the manuscript. 

Based on the foregoing, I consider that this manuscript can be published in Remote Sensing after this small correction.

Author Response

HYSPLIT acronym was given on line 182 (as we did not add it in the abstract where the term HYSPLIT is first introduced in our manuscript) and also in abbreviations list. We moved the abbreviation explanation in the abstract and removed it from line 182.

Reviewer 4 Report

The paper is focused on a study of a near real time automatic detection method, based on the synergy of continuous measurements taken by ceilometer and photometer, in order to detect lofted atmospheric aerosol layers and estimate aerosol load. The manuscript is well structured and good presented. The case study presented in this paper is interesting and clearly shows the weakness of the method. Indeed, the paper has some limitations due to the fact that the choice of a predefined threshold of 2500 m for the free troposphere layers is not correctly justify and induces errors in the study. As mentioned by the authors, ‘this could impact in missing some layers below 2500 m when the PBLH is significantly lower, especially during winter time’. Despite the fact that the authors evaluate the impact of the predefined threshold on the results by an off-line process to calculate the PBLH, I recommend the authors to revised the whole study. I suggest the authors to reprocess the data (off line), over the three-year periods, by considering the correct PBLH, to perform Hysplit backtrajectories and to determine the contribution of the aerosol pollution (layers’ air mass source, following Radenz et al), in order to enhance the quality of this paper. Moreover, the authors should consider the method to determine the PBLH (strongest negative gradient method) which may involve impacts on the final results and especially when the SNR is relatively low. I advise the authors to properly evaluate this method with the wavelet method developed in the frame of the EARLINET-ACTRIS activities. Finally, I would suggest to consider the paper for publication after a major revision. Indeed, this paper should be more enhanced for a successful scientific paper.

Author Response

We thank the new reviewer (Rev 4) for his careful reading and useful suggestion for the improvement of the study. A key element of his critique is that the approach to calculate PBLH (strongest negative gradient method) could induce major errors and that we should consider the wavelet method as a better candidate for our application. We fully agree with his suggestion.  Indeed, we already used the wavelet method as it was described in Wang et al. (2020) to calculate the PBLH; by mistake we provided a wrong citation that mixed the reviewer. We would like to thank him for bringing our attention to this issue and apologize for the confusion created. We have now replaced the citation of Sokol with Wang at al. (2020).

  1. Wang, D., Stachlewska, I.S., Song, X., Heese, B. and Nemuc A., Variability of the BoundaryLayer Over an Urban Continental Site Based on 10 Years of Active Remote Sensing Observations in Warsaw, Remote Sens., 12, 340, https://doi.org/10.3390/rs12020340, 2020

The reviewer further suggests revising the whole study reprocessing the data based on PBLH assessed off-line and identifying the missing layers. We would like to argue that his suggestion better fits a future study as they would significantly alter the scope of the current manuscript. We acknowledge that our approach has some limitations, which we discuss in our manuscript and try to determine their impact on the overall quality of our method. Especially, for the impact of the fixed threshold for the PBLH we dedicated a big part of section 3.4, where we provide an estimation of the missing layers due to the fixed height of 2.5 km. Reprocessing the data would significantly alter the main objective of our research, to present a methodology for the NRT detection of free tropospheric aerosol layers and their further closer monitoring. The off-line reprocess could alter to some extend the statistics presented considering the aerosol typing, but it would have no value for improving the proposed methodology. At a practical level, at this moment we are not able to reprocess all the three years considering PBLH already determined off-line. This reprocess takes quite a considerable amount of time, especially for the new runs of HYSPLIT. As mentioned, in the near future, several improvements are considered: PBLH determined in NRT (currently, several methods are under investigation, including STRATfinder), the use of an in-house algorithm to determine the ALH in the Free Troposphere (a first version was presented during the latest ELC2021: https://granada-en.congresoseci.com/elc2021/elc2021_proceedings_book_final!), while we consider the use of an ensemble backtrajectories in order to assess the air mass sources.

We have added the following paragraph at the end of Section 2.4, when discussing the limitations:

The fixed threshold of the 2.5 km, in some cases is above the PBLH. This could result to missing some layers between the PBLH and 2.5 km. This issue is further discussed in Section 3.4, where we try to estimate the impact of the fixed height threshold at the number of possible layers that have not been identified by the algorithm.

Reviewer 5 Report

I am happy with all the revisions and do not hesitate to recommend accepting the manuscript for publication. 

Author Response

We would like to thank Reviewer 5 for his previous review that helped us to improve our manuscript and for supporting its publication.

Round 2

Reviewer 4 Report

My questions have been satisfactorily answered. I believe that it is now acceptable in the present form.

This manuscript is a resubmission of an earlier submission. The following is a list of the peer review reports and author responses from that submission.

Round 1

Reviewer 1 Report

Comments on the “Towards early detection of tropospheric aerosol layers using monitoring with ceilometer, photometer and air-mass trajectories” by M. Adam et al.

A detection system has been developed based on ceilometer and photometer observations and is able to alert scientists who are doing lidar monitoring automatically. The manuscript is well written, the observation data is most advanced, the methodology is well developed, and the results are well presented and discussed. I believe the manuscript is suitable for publication as long as the following minor issues have been solved.

Minor issues:

  1. The abstract needs to be extended. It is important to introduce not only the detection system but also the climatology of the aerosol optical characteristics.
  2. Use “AERONET” or “Aeronet” over the entire manuscript.
  3. What is the definition of sea and ocean for aerosol source region, and what is the difference between “water” and “marine” source region types? Please clarity.
  4. L431, MERRA-2 is an analysis dataset, not a model.

Reviewer 2 Report

The paper describes the  complex for measuring the spatiotemporal aerosol distribution over a very large territory from Iberia to the Goby desert and presents the results obtained using this complex.

I have no major comments on the manuscript. Among the minor comments, are the following.

1. The altitudes in the figures are given above the ground lrvel. It would be better to give the altitude above the sea level to unify the altitude scale.s

 2. I recommend to delet a single quote from the capture of Fig.5 and explain this parameter in the text.

Among the advantages of the manuscript, the List of Acronims should be mentioned.

Reviewer 3 Report

Summary

The objective of this work is to implement an automatic detection system in near real time (NRT) for lofted aerosols layer, using a ceilometer located in Magurele-Romania. If the said instrument detects a layer of aerosols above 2500m, its optical properties during the event are compared with the optical properties of (almost) the last 10 years recorded by a solar photometer. In the case of significant deviations, an alert is sent to the scientists involved and the back trajectories are estimated (HYSPLIT model) to study the possible origin of the detected layer. Given lidars (a more precise technique, but high operating and maintenance costs) are commonly used for this task, the strength of this work is founded in using ceilometric measurements with the advantage of lower costs. However, it is important to consider the limitations of this type of measurement (low signal-to-noise ratio, temperature interferences, etc.).

General comments

The manuscript is clear and well-structured which makes it easy to follow and read. The title is a bit long, but correctly communicates the aim of the work. The abstract is concrete, but I think it would be beneficial to include a summary of the results (for example the effectiveness of the NRT detection system).

Concerning the introduction, although it gives a brief description of the problem, it does not make clear what is the importance of an early warning system related to aerosols in the free troposphere. It would be paramount to justify the reason for choosing to monitor the free troposphere with an NRT system, when the most important problem (in terms of impacts on human activity) occurs in the PBL (otherwise it would be desirable to communicate its possible uses).

According to Cartea et al. (2019) “The RADO site is mainly influenced by the city emissions from March to November. These types of aerosols are mainly in the lower part of the atmosphere, within the planetary boundary layer (PBL)”. While the Aeronet measurements in Magurele are being mainly influenced by local emissions, the ceilometer (located in the same place as the solar photometer) observes a more regional characteristic of the aerosols (heights greater than the PBL), so the use of both data sources to define an alert is misleading.

The latter is directly connected with the proposed methodology, which lacks an analysis that couples early warnings with local/regional air pollution events (or other undesirable events), helping in that way to understand/communicate the effectiveness of the early warning. If we put ourselves in the role of user or decision maker, we would be interested in knowing what the reliability of a system is. How many of the triggered alerts (around 5000 in 3 years) correspond to the prevention of events of interest to users/decision makers? Furthermore, from the number of alerts in the last three years, it seems (although no data is found in the text) indicating how many of them are part of the same event (a continuation of previous alerts).

The methodology also does not indicate whether or not there are quality controls of the ceilometer data (and how they are carried out), just as it does not present (if it exists) an in-situ validation of the instrument's capabilities. Although other articles that compare/validate the results of ceilometer measurements with lidars are mentioned, its low signal-to-noise ratio makes the in-situ validation a necessary requirement.

The use of optical properties monthly means as a limit to trigger or not the alert is striking, without first verifying the effectiveness of these limits. From the article reading, it can be deduced that many of the alerts generated could be avoided (false positives) if a metric to be used based on the particular problem is designed and thus make a tighter decision (staying on the safe side to limit false negatives, that would be more concerning).

Finally, and despite the aforementioned methodological flaws, I think the manuscript has great potential for addressing/preventing pollution events. If the methodology is improved, I think this article will bring a novel contribution with a positive impact on the community.

Specific comments:

-Line 61: it is suggested to start a new paragraph to describe the NRT systems

-Line 105 says: “One secondary goal of this study is to exploit the information provided by the ceilometer, as given in the raw data”. It is not clear from the rest of the text how this was performed.

-Line 171: “One backward trajectory is calculated for each detected aerosol layer (a.s.l.) for a backward run time of 240 h.” What are the criteria for choosing 240hs?

-Line 204: It is not understood if the comparisons are made between the suspicious data and the distribution for that particular month in which the suspicious data was measured (10 data points to build the CL for January, 10 data to build the CL for January February, …etc.). Or is it that the suspicious data is compared with every month in the 10 years (120 data points to build the confidence limits)?

-Figure 2: Although it is understandable that these plots are part of the “warning” email sent to the interested parties, it is suggested to improve the visual quality to allow the reader to interpret it more easily. Also, the footnote incorrectly identifies plots b) and c). What does it mean “RMSE” in this context?

-Line 314 to 336 seems to be more proper for the methodology section than for results and discussions

-Line 487. What “FT” stands for?

-I suggest moving Fig 5 y 7 to the appendix as the distract the reader from the main point (i.e. the early warning)

-Line 87: “These kinds of NRT alert systems are not limited only on pollution assessment, but they can be used for any other environmental hazard.”

-Line 164: “Level 1.5 data are cloud-screened and quality controlled, but not quality-assured as level 2 data, since the two consecutive calibrations of the instrument have not been verified.”

-Table 1: 2 identical typos in Percentiles

Reviewer 4 Report

The study describes a methodology for the early detection of heavy pollution events in the troposphere by means of the integrated use of a ceilometer and a photometer. The automated system allows to evaluate the degree of pollution based on photometer measurements but, as the authors themselves indicate as a limitation of the current version of their procedure: “the degree of pollution is not established in most cases due to the lack of radiometric data”. In my opinion this aspect can’t be improved in the future and for this reason I suggest to reject the paper in its present form.

My suggestion for a new submission is to change the objective of the study into a methodology to promptly study high intensity pollution events in the troposphere.  At the same time, authors could try to improve their automated procedure following for example Papagiannopoulos, N. et al., 2020.

Reference

Papagiannopoulos, N., D'Amico, G., Gialitaki, A., Ajtai, N., Alados-Arboledas, L., Amodeo, A., Amiridis, V., Baars, H., Balis, D., Binietoglou, I., Comerón, A., Dionisi, D., Falconieri, A., Fréville, P., Kampouri, A., Mattis, I., Mijić, Z., Molero, F., Papayannis, A., Pappalardo, G., Rodríguez-Gómez, A., Solomos, S., and Mona, L. An EARLINET early warning system for atmospheric aerosol aviation hazards, Atmos. Chem. Phys., 2020, 20, 10775–10789, https://doi.org/10.5194/acp-20-10775-2020